# Fantastic Reasoning Behaviors and Where to Find Them: Unsupervised Discovery of the Reasoning Process

**Zhenyu Zhang**[1]   **Shujian Zhang**[2]   **John Lambert**[2]   **Wenxuan Zhou**[2]   **Zhangyang Wang**[1]   **Mingqing Chen**[2]
**Andrew Hard**[2]   **Rajiv Mathews**[2]   **Lun Wang**[2]

## Abstract

Despite the growing reasoning capabilities of recent large language models (LLMs), their internal mechanisms during the reasoning process remain underexplored. Prior approaches often rely on human-defined concepts (e.g., overthinking, reflection) at the word level to analyze reasoning in a supervised manner. However, such methods are limited, as it is infeasible to capture the full spectrum of potential reasoning behaviors, many of which are difficult to define in token space. In this work, we propose an unsupervised framework (*namely*, RISE: **R**easoning behavior **I**nterpretability via **S**parse auto-**E**ncoder) for discovering *reasoning vectors*, which we define as directions in the activation space that encode distinct reasoning behaviors. By segmenting chain-of-thought traces into sentence-level 'steps' and training sparse auto-encoders (SAEs) on step-level activations, we uncover disentangled features corresponding to interpretable behaviors such as reflection and backtracking. Visualization and clustering analyses show that these behaviors occupy separable regions in the decoder column space. Moreover, targeted interventions on SAE-derived vectors can controllably amplify or suppress specific reasoning behaviors, altering inference trajectories without retraining. More interestingly, SAEs enable the discovery of novel behaviors beyond human supervision. We demonstrate the ability to control response confidence by identifying confidence-related vectors in the SAE. These findings underscore the potential of unsupervised latent discovery for both interpreting and controllably steering reasoning in LLMs.

[1]The University of Texas at Austin [2]Google DeepMind. Correspondence to: Zhenyu Zhang <zhenyu.zhang@utexas.edu>.

*Proceedings of the 43$^{rd}$ International Conference on Machine Learning*, Seoul, South Korea. PMLR 306, 2026. Copyright 2026 by the author(s).

## 1. Introduction

Recent advancements in reasoning have significantly expanded the capabilities of large language models (LLMs), moving them far beyond basic language understanding to encompass more complex reasoning tasks. These include competition-level mathematical problem solving (Ahn et al., 2024), project-level coding (Jiang et al., 2024), and planning (Huang et al., 2024; Valmeekam et al., 2023). Evidence from model responses suggests that such substantial performance improvements primarily stem from their ability to perform extended chain-of-thought (CoT) reasoning (Wei et al., 2022). Nevertheless, how such lengthy reasoning trajectories contribute to performance remains underexplored.

Recent studies have investigated the reasoning process, such as analyzing entropy mechanisms during inference (Fu et al., 2025; Zhang et al., 2025) or at the post-training stage (Cui et al., 2025; Zhao et al., 2025; Wang et al., 2025b), which can be further leveraged to achieve more accurate and concise reasoning (Yang et al., 2025). Other works suggest that current reasoning processes are often verbose, with certain parts being ineffective (Fu et al., 2026; Huang et al., 2025; Sheng et al., 2025). In addition, another line of studies argues that reasoning ability is tied to specific behaviors (Chen et al., 2025; Venhoff et al., 2025; Ward et al., 2025; Gandhi et al., 2025), such as *reflection* (*i.e.*, the model revisits and verifies its previous reasoning steps) and *backtracking* (*i.e.*, the model abandons the current reasoning path and pursues an alternative solution), which appear to be closely associated with the achieved performance gains.

From the perspective of mechanistic interpretability, many studies rely on activation engineering methods, particularly the Difference-of-Means (DiffMean) approach (Marks & Tegmark, 2023). This method begins by constructing a contrastive dataset with human supervision, such as categorizing samples into safe vs. harmful or positive vs. negative groups. For each category, it then extracts the latent representations of individual samples (*i.e.*, $h_i^+$ and $h_j^-$) and computes a steering vector by averaging the representations within each category and taking their difference: $v = \frac{1}{N_+} \sum_{i=1}^{N_+} h_i^+ - \frac{1}{N_-} \sum_{j=1}^{N_-} h_j^-$. The resulting steering vector can then be applied to shift the response style, for

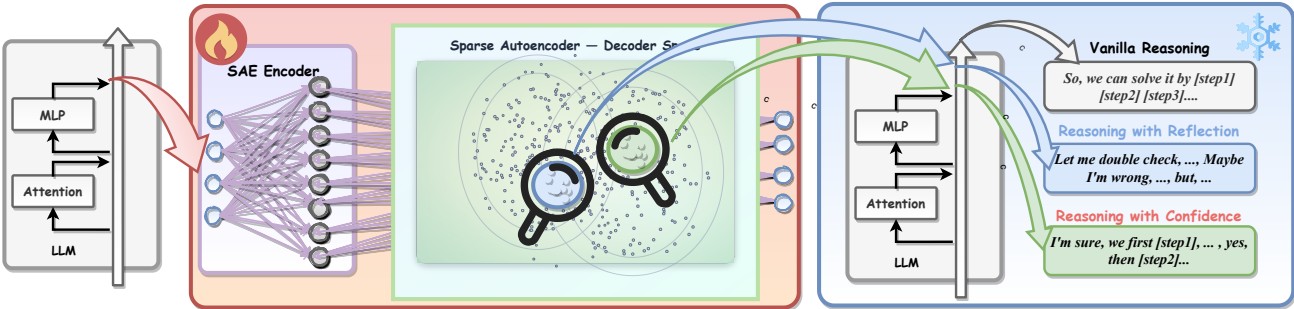

*Figure 1.* Illustration of our `RISE` framework for unsupervised reasoning behavior discovery. The pipeline consists of two stages: (i) training a Sparse Autoencoder (SAE) on unlabeled representations of reasoning steps (Left), and (ii) evaluating causal effects on the original reasoning process (Right). Notably, the intervention process on the right is applied directly, without any additional training.

example, from sad to happy or vice versa.

However, such activation-based approaches are ill-suited for understanding reasoning, as they rely on supervised, pre-defined concepts. This is feasible in tasks like sentiment classification (Pang et al., 2002), where concepts such as "happy" and "sad" are clearly separable. In contrast, reasoning behaviors are fluid, overlapping, and difficult to annotate at scale, which has led prior studies to focus narrowly on a small set of predefined behaviors such as reflection or backtracking (Wang et al., 2025d). This restricts the scope of analysis and the robustness of the resulting steering vectors.

In this work, we address these challenges by introducing an unsupervised framework (*i.e.*, `RISE`: **R**easoning behavior **I**nterpretability via **S**parse auto-**E**ncoder) for discovering reasoning behaviors. Building on the linear representation hypothesis (Park et al., 2023), we define fine-grained reasoning behaviors as linear directions in the activation space, which we refer to as *Reasoning Vectors*. These vectors can be applied to modify the reasoning mechanism in specific ways and thereby influence response quality.

Unlike prior approaches, our framework does not rely on human-supervised labels; instead, it directly identifies reasoning vectors from step-level activations in chain-of-thought sequences. To achieve this, we employ sparse autoencoders (SAEs), which learn a dictionary of sparse latent features that reconstruct hidden states while promoting disentanglement. We show that individual decoder columns correspond to interpretable reasoning behaviors (Figure 1). Visualizations further reveal that these vectors cluster into semantically coherent regions of activation space. Most importantly, we demonstrate *controllability*: injecting SAE-derived vectors during inference can suppress or amplify behaviors such as reflection, directly altering the reasoning trajectory without additional training. Beyond semantic behaviors, we also observe evidence of *structural organization*: decoder columns consistently form clusters aligned with response length, with separability peaking in mid-to-late layers. Furthermore, we show that SAEs can reveal

*new behaviors* that are difficult to define through word-level human supervision. As a case study, we examine the semantic concept of confidence and find that confidence-related directions are highly concentrated, with interventions on the corresponding vectors exerting clear causal effects on reasoning styles. Our contributions are threefold:

- We propose an unsupervised framework, *i.e.*, `RISE`, that captures the structure of reasoning behaviors within the latent space. Unlike prior methods that rely on human-defined concepts, `RISE` models diverse reasoning behaviors in an end-to-end fashion directly within the decoder column space.

- With `RISE`, we show that SAE-derived vectors align with human-interpretable behaviors and can be used to selectively modulate reasoning at inference time. In particular, we can directly control reflection and backtracking during reasoning by reducing or enhancing the components of hidden representations along the SAE column directions on the fly.

- We further demonstrate the ability to discover novel reasoning behaviors that are difficult to define with word-level human supervision. As an example, we consider *confidence*, a behavior that is challenging to specify at the word level. We find that confidence-related reasoning vectors form coherent clusters and causally shift the model's response style toward more confident answers.

## 2. Related Works

**Reasoning Models.** Previous works have demonstrated that enabling models to generate longer outputs can significantly enhance their reasoning ability. This progress starts from the famous Chain-of-Thought (CoT) prompting (Wei et al., 2022), to test-time scaling (Snell et al., 2024), and more recent reinforcement learning–optimized reasoning models, such as OpenAI's *o*-series (Jaech et al., 2024), Anthropic's *Claude-3.7-Sonnet-Thinking* (Anthropic, 2025),

and Google's *Gemini-2.5-Flash* (Google, 2025), as well as notable open-source counterparts (Guo et al., 2025; Yang et al., 2024; Team, 2024; Abdin et al., 2025; Team, 2025). While longer responses generally improve performance, the mechanisms remain unclear. Recent studies link these gains to several metacognitive behaviors (Gandhi et al., 2025; Chen et al., 2025; Venhoff et al., 2025), yet they rely on human-defined behaviors. In this work, we seek to uncover the geometry of reasoning behaviors in an unsupervised manner, thereby avoiding dependence on human labeling.

**Activation Steering.** Activation steering modifies model outputs by editing internal activations, with notable approaches including representation engineering (Zou et al., 2023), activation patching (Meng et al., 2022), and DiffMean (Marks & Tegmark, 2023). These methods have been effective in domains such as improving LLM truthfulness (Wang et al., 2025c), enhancing privacy (Goel et al., 2025), and controlling sentiment (Han et al., 2023). However, they typically rely on contrastive pairs (e.g., happy vs. sad) to define steering directions, which is suitable when clear oppositional concepts exist. In reasoning, such contrasts are more difficult to define: prior work has targeted specific behaviors like reflection (Chen et al., 2025; Venhoff et al., 2025) or coarse distinctions between short and long responses (Huang et al., 2025; Sheng et al., 2025; Eisenstadt et al., 2025), leaving the broader reasoning behavior space underexplored. In this work, we present a pivot study to investigate this space in an unsupervised manner. A recent effort also explored reasoning models in an unsupervised way by using a Sparse Auto-Encoder (SAE) (Wang et al., 2025a), but primarily treated SAE as a bridge to transfer reasoning ability from a reasoning model to a base model via supervised fine-tuning, offering limited insight into the internal mechanisms of the reasoning process. Our work instead aims to directly analyze these mechanisms.

## 3. Preliminary

### 3.1. Sparse Auto-Encoder

Building on the Linear Representation Hypothesis (Park et al., 2023), each atomic reasoning behavior can be mapped to a specific direction in the activation space, which we define as a reasoning vector. To automatically identify such behaviors, we employ sparse auto-encoders (SAEs), which provide an effective and principled approach. An SAE consists of an encoder and a decoder that aim to reconstruct representations in the activation space:

$$\hat{h} = W_{\text{decoder}}^\top \sigma(z) + b_{\text{decoder}}; z = \sigma\big(W_{\text{encoder}}^\top h + b_{\text{encoder}}\big) \tag{1}$$

Here, $h \in \mathbb{R}^d$ denotes the original representation, $\sigma$ is a non-linear activation function, and $\hat{h}$ is the reconstruction. Each row of $W_{\text{decoder}} \in \mathbb{R}^{D \times d}$ corresponds to an atomic vector

(*i.e.*, a reasoning vector), where $D$ is the hidden dimension of the SAE and $d$ is the dimension of the original input. The latent feature $z$ represents the corresponding code. For a standard SAE (Cunningham et al., 2023), we use ReLU as the non-linear activation function $\sigma$. The training objective is formulated as

$$\mathcal{L} = \|\hat{h} - h\|_2^2 + \lambda \|z\|_1, \tag{2}$$

which encourages the SAE to accurately reconstruct the original activations while enforcing sparsity in the latent codes. This ensures that only a small number of atomic concepts are used, thereby reducing entanglement across vectors and enhancing interpretability.

### 3.2. Thought Representation Construction

Intuitively, reasoning behaviors cannot be trivially explained by token-level attributes, since the same token can play diverse roles across different contexts. This observation, further supported by recent studies (Ward et al., 2025; Venhoff et al., 2025), motivates our choice of analyzing the reasoning process at the sentence level, where the internal structure of the whole responses can be more meaningfully captured.

The construction of thought representations for SAE training involves the following steps: (i) *Collecting model responses:* we feed each question from the selected training set into the target model to generate responses. (ii) *Splitting responses into sentence-level steps:* we divide each response into reasoning steps using the delimiter symbol `<\n\n>`, producing $k$ steps per response, where $k$ varies across samples. (iii) *Embedding each step:* we re-run inference by feeding both the question and the corresponding response into the model, and extract the hidden representations of the token `<\n\n>`. Each representation is then regarded as the activation of its corresponding reasoning step (Chen et al., 2025). The representations are then concatenated, denoted as $\{h_i^l\}$, where $i \in \{1, \ldots, N\}$ indexes the samples and $l$ denotes the layer index. We specifically use the residual stream representations after each transformer layer, and train the SAE on a single chosen layer.

## 4. Unsupervised Reasoning Vector Discovery

### 4.1. SAE Discovers Reasoning Behaviors in the Decoder Column Space

To begin, we first analyze the ability of the SAE to capture the ground-truth representation structure. For each activation $h$ obtained from the original model, the SAE decoder reconstructs it as: $\hat{h} = W_{\text{decoder}}^\top \sigma(z) + b_{\text{decoder}}$. This reconstruction can be interpreted as a linear combination of selected rows of $W_{\text{decoder}}$. Under the linear representation hypothesis, we posit that the true activation admits a sparse representation over a ground-truth dictionary

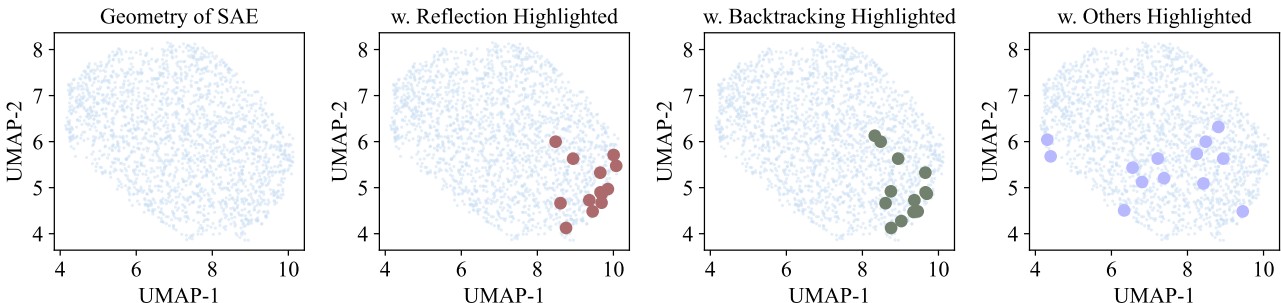

*Figure 2.* Visualization of SAE decoder columns projected onto a 2-D plane with UMAP. We show the raw SAE decoder rows and the corresponding results with human-defined behaviors highlighted. Results are obtained from the final layer of R1-1.5B.

$W = [w_1, \ldots, w_m] \in \mathbb{R}^{d \times m}$. As established in Theorem 4.1, the SAE decoder matrix can recover this dictionary in the decoder space up to additional permutation and scaling. More details can be found in Appendices A and B.

**Theorem 4.1.** *Suppose hidden representations at delimiter tokens follow the generative model*

$$h = Wa + \varepsilon, \qquad (3)$$

*where $h \in \mathbb{R}^d$, $W = [w_1, \ldots, w_m] \in \mathbb{R}^{d \times m}$ is a dictionary of latent behavior directions, $a$ is a $k$-sparse code, and $\varepsilon$ is bounded noise. Assume: (i) Incoherence: $\max_{i \neq j} \frac{|\langle w_i, w_j \rangle|}{\|w_i\| \|w_j\|} \leq \mu < 1$. (ii) Sparsity: $k < c/\mu$ for a universal constant $c$. (iii) Activation: SAE uses ReLU nonlinearity. (iv) Separation: Nonzero coefficients satisfy $|a_i| \geq \alpha > 0$.*

*Then, as the number of samples $N \to \infty$, any local optimum of the SAE training objective*

$$\min_{W_{enc}, W_{dec}} \mathbb{E}\big[\|h - \hat{h}\|_2^2\big] + \lambda \|z\|_0 \qquad (4)$$

*recovers a decoder matrix whose columns align with the true dictionary up to permutation matrix $\Pi$ and scaling diagonal matrix $D$:*

$$\exists \Pi, \ D \succ 0 \ \ s.t. \ \ W_{dec} \approx W\Pi D. \qquad (5)$$

### 4.2. Setup Details

We use DeepSeek-R1-Distill-Qwen-1.5B (R1-1.5B) (Guo et al., 2025), along with 500 randomly sampled training examples from the MATH dataset (Lightman et al., 2023). We follow Section 3.2 to obtain the activations for SAE training. Since our goal is to model various reasoning behaviors, whose complexity is much lower than modeling raw language structure, we adopt a relatively small hidden dimension of $D = 2048$. The SAE is trained with a batch size of 1024 and a learning rate of $1 \times 10^{-4}$, with a warm-up over the initial 10% of training. We use the Adam optimizer (Kingma, 2014) with cosine annealing learning rate decay. To encourage sparsity, we apply a sparsity strength

of $\lambda = 2 \times 10^{-3}$. Additionally, we extend our analysis to Qwen3-8B and report the results in Figure 6.

### 4.3. Visualizing the Geometry of the SAE Decoder

We then conduct a human-supervised analysis of the SAE decoder geometry to assess whether the structures learned by the SAE align with human supervision. Specifically, we classify each representation in $\{h_i^l\}$ into one of three categories: *reflection*, *backtracking*, or *other*. The main motivation for emphasizing reflection and backtracking is that, compared to prior chain-of-thought prompting, current reasoning models introduce stages of answer refinement through these processes, which are the most prominent features we aim to explore.

The classification is performed using an LLM-as-a-judge approach, where for each reasoning step associated with $h_i^l$, we prompt the LLM (*i.e.*, GPT-5) with precise definitions of the behaviors: *reflection* (re-examining earlier steps), *backtracking* (switching to a new approach), and *other*. For each labeled step, the corresponding representation is then mapped back to the SAE latent feature space.

More details on the annotation process are provided in Appendix C. To analyze the robustness of the annotation method, we compare annotation results produced by different LLM judges and a keyword-matching approach, using the classification criteria defined in Table 3. The results, shown in Figure 3, report the agreement ratio (defined as the proportion of steps receiving the same annotation relative to all steps) exceeding 85% for each pair of methods. Overall, the consistency is high across most comparisons, with GPT-5 and GPT-4o exhibiting particularly strong agreement at approximately 94%. These results validate the reliability and consistency of our annotation methodology.

We record the activation patterns and report the top-active channels, highlighting their associated decoder columns. Here, the activity of a channel is measured by the largest magnitude of its latent feature. Additionally, we apply Uniform Manifold Approximation and Projection

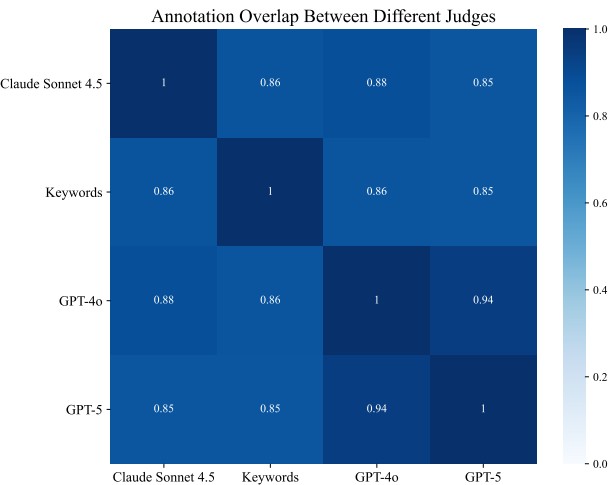

*Figure 3.* Results of annotation consistency across different methods. The numbers represent the agreement ratio for each pair of annotation methods.

(UMAP) (McInnes et al., 2018) to embed the decoder columns into a two-dimensional space.

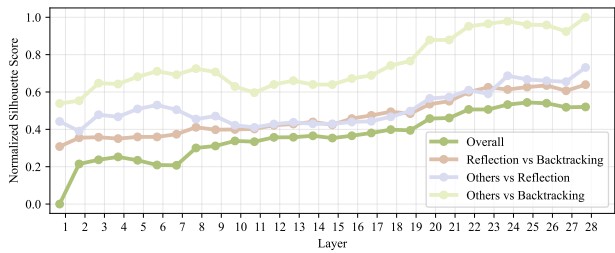

*Figure 4.* Results of normalized Silhouette scores across different layers of R1-1.5B.

**Results**. Figure 2 illustrates that the SAE decoder columns encode semantically meaningful behaviors in the latent space. The leftmost subfigure shows the UMAP projection of all decoder vectors without annotations. When overlaid with human-labeled behavioral concepts, clear semantic structures emerge. We adopt UMAP for visualization because it leverages cosine similarity as the internal metric, emphasizing directional rather than Euclidean differences. This choice is particularly appropriate since activation vectors are primarily meaningful in terms of their direction rather than their magnitude. Furthermore, for behaviors such as reflection and backtracking, which are semantically clearer from a human perspective, the corresponding decoder columns cluster more tightly in localized regions. In contrast, for the *other* category, which potentially encompasses a mixture of behaviors, the top-active vectors are less centered and more dispersed across the entire SAE space.

**Layer-wise Properties of SAE Geometry**. Then, we quantify the above analysis by Silhouette scores (Rousseeuw,

1987; Shahapure & Nicholas, 2020). This metric measures both the cohesion of samples within a cluster and their separation from other clusters. We normalize the raw scores across layers for better visualization. The results are shown in Figure 4. Two key observations can be made. First, later layers generally achieve higher Silhouette scores than earlier ones, indicating that behavioral concepts become more separable as representations deepen. Interestingly, a slight decline occurs near the final layers (except for the very last one), suggesting that mid-to-late representations encode the most distinct behavioral structures. This observation aligns with the oversmoothing phenomenon in current LLMs (Wang et al., 2023), where token representations become excessively similar. Second, the comparison between *reflection* and *backtracking* reveals only modest differences, whereas the separation of "other" behaviors from either reflection or backtracking is substantially stronger. This suggests that reflection and backtracking occupy more overlapping representational subspaces, while both are more clearly distinguished from the residual "other" category.

### 4.4. Causal Examination of SAE Decoder Columns

#### 4.4.1. INCORPORATING SAE COLUMNS DURING LLM INFERENCE

**Incorporating SAE Columns during LLM Inference**. We next perform an intervention study to examine the causal effect of SAE decoder columns, focusing on those from the final layer. We first filter out decoder columns that exhibit strong activations across multiple behaviors (*e.g.*, reflection and backtracking). From the remaining reflection-specific columns, we compute their average to obtain a single *reflection vector*. During inference, this vector is injected into the hidden representation of the last token at each reasoning step, as shown in Figure 5. Note that all decoder columns are independently normalized to unit length $\|w_i\|_2 = 1$. The intervention is then performed as:

$$h' = h - w_i(w_i^\top h), \qquad (6)$$

where $w_i$ denotes decoder column $i$. In this way, we project out the component of $h$ along the reflection direction to examine how the reflection behavior changes in the model's responses.

**Intervention Results**. As shown in Figure 7, when DeepSeek-R1-1.5B is prompted with a detailed math question, its reasoning style can be effectively shaped while preserving correctness. Across all conditions, the model consistently produces the same final answer $(3, \pi/2)$, yet the reasoning trajectory shifts in a predictable manner: negative intervention suppresses meta-cognitive phrases (*i.e.*, reflection) and shortens the whole response process, positive intervention amplifies self-checking behaviors while lengthening it, and vanilla inference lies between. While

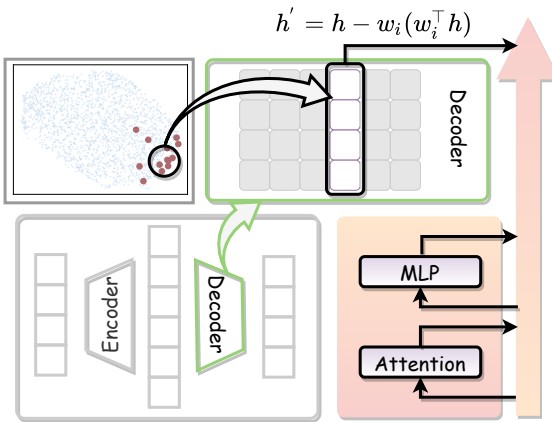

$$h' = h - w_i(w_i^\top h)$$

*Figure 5.* Illustration of our inference process that utilizes SAE decoder columns. For a given reasoning behavior, we compute the corresponding centroid in the SAE decoder column space and directly apply it during inference of the original model to examine how the response changes.

a simple example may appear trivial, we further validate the causal effect of SAE columns across diverse tasks, behaviors, and models. As shown in Figure 6, the results consistently demonstrate the causal influence of SAE columns. Specifically, applying negative interventions leads to a consistent decline in the corresponding behavior, whereas positive interventions result in clear enhancement. Moreover, when we manually control the intervention strength with a scalar value as: $h' = h - \alpha \cdot w_i(w_i^\top h)$, where $\alpha \in \{-1.5, -1, 0, 1, 1.5\}$, the corresponding number of reflection steps of DeepSeek-R1-1.5B on the AIME25 tasks changes to $\{58.6, 73.6, 90.5, 131.0, 166.9\}$, consistently evolving in accordance with the intervention strength. These results further support that SAEs learn meaningful reasoning behaviors in an unsupervised manner.

**Generalization Across Data Domains**. The SAE is trained on the MATH500 dataset, after which we evaluate how the learned vectors generalize to other domains. To ensure a larger domain shift, we consider commonsense and logical reasoning tasks, specifically GPQA-Diamond (Rein et al., 2023) and KnowLogic (Zhan et al., 2025). Notably, we apply the same SAE columns during inference on these new domains with the R1-1.5B model. The results, reported in Table 1, show that the reflection and backtracking vectors consistently modulate reasoning behaviors across domains.

**SAE Reveals the Underlying Geometry of Response Length.** Inspired by recent findings (Huang et al., 2025; Sun et al., 2025) that response length can be distinguished in the activation space, we are further motivated to explore the geometry of response length in the SAE decoder space. Using the same procedure, the results are presented in Figures 11 and 12. We observe that early layers exhibit weak separation with respect to response length, whereas mid-

to-late layers show stronger alignment with this structural property. More details can be found at Appendix D.

### 4.5. Discovering New Behaviors with SAE

In the previous section, we demonstrated that an unsupervisedly trained SAE can capture semantically meaningful structures, as evidenced by visualizing its weight geometry and its alignment with human-defined reasoning behaviors. In the following, we present an initial attempt showing that SAEs can also be used to discover new behaviors.

From the decoder columns of the SAE, the key challenge becomes selecting a specific subspace that can be leveraged for a particular goal. As a starting point, we consider identifying a goal that meaningfully impacts the reasoning process. Recent studies on the entropy mechanisms of LLM reasoning (Fu et al., 2025; Zhang et al., 2025; Cui et al., 2025; Zhao et al., 2025; Wang et al., 2025b) highlight that entropy, or equivalently model confidence, plays a critical role during the reinforcement stage. For instance, incentivizing an LLM to maximize its confidence in the final answer has been shown to positively influence reasoning ability in certain scenarios. Moreover, confidence can also serve as a criterion for filtering out ineffective responses (Yang et al., 2025; Zhang et al., 2025; Fu et al., 2025), thereby improving inference efficiency. Motivated by these insights, we adopt entropy as our proxy objective and seek to identify decoder columns that are closely associated with entropy.

**Setup.** Given a trained SAE with decoder $W_{\text{decoder}} \in \mathbb{R}^{D \times d}$, where $d$ denotes the dimension of the original model and $D$ is the number of learned vectors, we divide the original model into two parts. The splitting point is chosen at the layer $l$ from which the SAE is trained. Accordingly, we denote the remaining part of the model as $h = f_{1 \to l}(x); y = f_{l \to L}(h)$, where $L$ is the total number of layers, $x$ is the input token and $h$ is the activation at layer $l$.

**Identifying Target SAE Decoder Columns.** We assign a score vector $S \in \mathbb{R}^D$ to the $D$ decoder columns and optimize $S$ by minimizing the final entropy. Specifically, given the activation set $\{h_i^l\}$ extracted from the MATH500 training set at layer $l$, we solve the following objective:

$$\arg\min_S \mathbb{E}\left[-\sum_{k=1}^{|V|} p_k \log p_k\right], \tag{7}$$
$$\text{where } p_k = \text{softmax}\big(f_{l \to L}(h + SW_{\text{decoder}})\big)_k.$$

where $p_k$ denotes the predicted probability of token $k$ and $|V|$ is the vocabulary size. We solve this optimization using the Adam optimizer with a learning rate of 0.01 for 1000 iterations, together with a cosine annealing learning rate decay schedule. The batch size is 256.

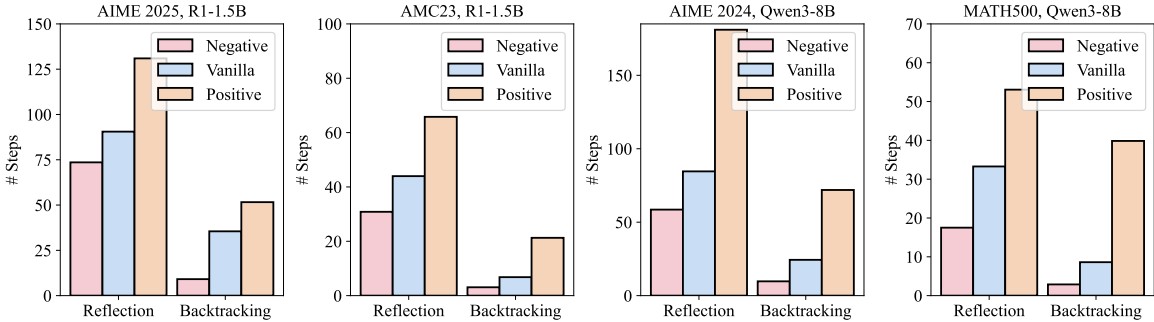

*Figure 6.* Statistics of reasoning behavior shifts induced by SAE column interventions are reported across different models and tasks, where the SAE columns are consistent across tasks.

*Table 1.* Generalization of SAE columns under domain shift. The SAE columns are learned from the MATH500 dataset and applied to other domains. Numbers denote the corresponding steps.

| Steering Direction | GPQA-Diamond | | KnowLogic | |
|---|---|---|---|---|
| | **Reflection** | **Backtracking** | **Reflection** | **Backtracking** |
| Vanilla | 53.23 | 11.83 | 35.56 | 5.42 |
| Positive | 62.77 | 20.31 | 51.00 | 9.38 |
| Negative | 45.42 | 6.47 | 25.99 | 2.33 |

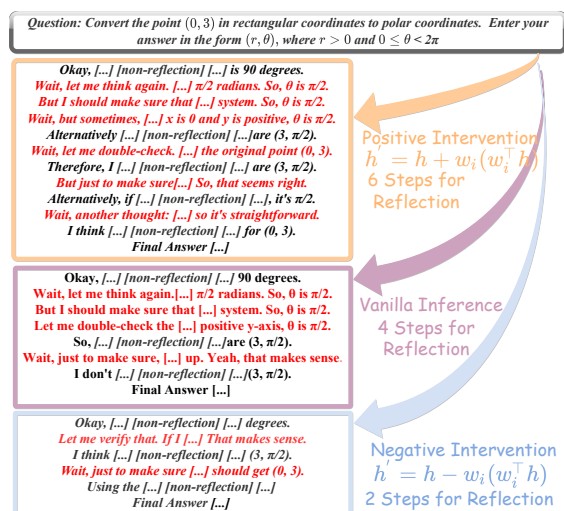

*Figure 7.* Responses from the R1-1.5B model when intervened with SAE vectors corresponding to reflection behaviors. Reasoning steps associated with reflection are highlighted in red. Zoom in for better visualization.

## 4.6. Emergence of Confident Reasoning Vectors

We conduct experiments on the last layer of DeepSeek-R1-1.5B. As shown in Figure 8, we select the columns with the highest scores in $S$, which implies that these columns play a critical role in achieving low entropy. We then inject with the corresponding vector, referred to as the *confidence reasoning vector*, during inference of the original model. The intervention process follows the same procedure described in the previous section 4.4.

**Confidence Vectors Concentrated in the SAE Space**. First, we observe that the confidence-related columns are predominantly located in the bottom-right region of the visualization, suggesting that this area is particularly important for reasoning confidence. Moreover, these columns partially overlap with the reflection and backtracking regions identified in Figure 2, indicating that reflection and backtracking behaviors may contribute to higher entropy. This observation is consistent with recent findings (Wang et al., 2025b). Further results are provided by Figures 8(b–c), where we visualize the most frequent tokens at the beginning of each reasoning step. Without intervention, frequent tokens include reflection-related cues such as "Wait" and backtracking indicators such as "Alternatively." After intervention with the confidence vector, these are replaced by tokens more closely related to detailed mathematical calculation steps.

**Causal Effect of Confidence Vectors**. In addition, we measure the frequency of reflection and backtracking steps before and after intervention. On the AIME25 tasks, reflection steps are reduced from 90.53 to 33.77, and backtracking steps decrease from 35.50 to 5.93. More importantly, the intervention substantially alters the reasoning style, with large reductions in reflection and backtracking. These results demonstrate that SAEs can be used not only to interpret but also to *actively modify* the reasoning style of LLMs. Furthermore, as shown in Table 2, we observe that the confidence vector also generalizes across domains, exhibiting a clear ability to simultaneously modulate both reflection

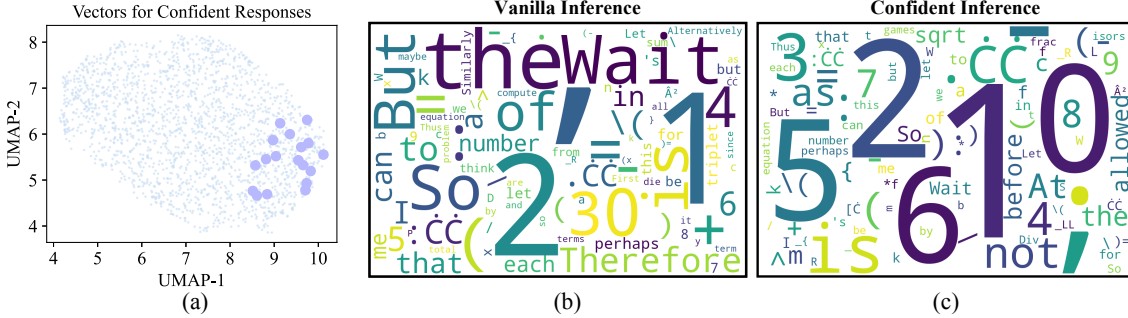

*Figure 8.* (a) Visualization of SAE decoder columns, with top-scoring columns related to entropy highlighted, showing clear clustering. (b) Frequent tokens during vanilla inference. (c) Frequent tokens under intervention with the entropy-related vector. Comparing (b) and (c), tokens associated with reflection or backtracking become less frequent (*e.g.*, Wait), while more tokens emerge that correspond to confident mathematical calculation (*e.g.*, numbers)

*Table 2.* Generalization of confidence vectors across domains with different steering directions. Numbers indicate the corresponding steps. The confidence vectors are derived from the MATH500 training set and applied to out-of-distribution commonsense tasks. The increase and decrease in steps demonstrates the causal effect of applying these vectors.

| Steering Direction | GPQA | | KnowLogic | |
|---|---|---|---|---|
| | **Reflection** | **Backtracking** | **Reflection** | **Backtracking** |
| Vanilla | 53.23 | 11.83 | 35.56 | 5.42 |
| Positive | 61.37 | 16.18 | 48.16 | 5.98 |
| Negative | 37.25 | 6.91 | 20.04 | 3.51 |

and backtracking behaviors. This further demonstrates the semantic similarity between the confidence direction and the combined effects of reflection and backtracking.

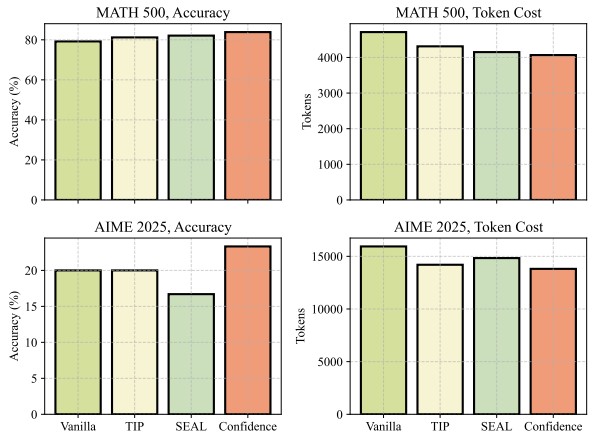

*Figure 9.* Results on reasoning accuracy and token cost under different steering methods on R1-1.5B.

## 5. Reasoning Enhancement via Confidence-Vector Intervention

We further explore potential applications of RISE for enhancing the reasoning process. Inspired by recent work (Hu et al., 2025) that learns a bias term at test time during the pre-filling stage, we select the top-3 confidence vectors $c_1, c_2, c_3$ and learn their combination coefficients at test time to con-

trol the reasoning process. As illustrated in Figure 5, this yields a sample-dependent steering vector $\sum_{i \in 1,2,3} \alpha_i c_i$. We compare this approach against SEAL (Chen et al., 2025) and TIP (Wang et al., 2025d). As shown in Figure 9, our careful steering strategy improves reasoning accuracy by up to 4.66 points while achieving a 13.69% reduction in token usage. These results serve as a pivotal case study demonstrating the potential of RISE to enable on-the-fly control of the reasoning process through interpretability.

## 6. Conclusions

We propose an unsupervised framework for discovering and controlling reasoning behaviors in LLMs. By training SAEs on chain-of-thought activations, we identify disentangled latent vectors corresponding to semantic reasoning behaviors. These results extend the linear representation hypothesis to reasoning, showing that abstract cognitive patterns can be linearly encoded and selectively manipulated. We further uncover a confidence-aware structure in the latent space, where latent directions aligned with response confidence can be identified, which is difficult to achieve directly in text space. Our findings offer new opportunities for interpretability and control: unsupervised discovery provides a scalable way to map reasoning behaviors without manual labels, controllable latent directions enable fine-grained test-time adaptation, motivating future work on aligning structural and semantic features for more reliable reasoning.

## Impact Statement

This work presents an unsupervised approach for discovering and controlling reasoning behaviors in large language models at test time, aiming to improve interpretability. By identifying internal representations associated with behaviors such as reflection and backtracking and enabling targeted interventions, the method helps diagnose and mitigate inefficient or unstable reasoning, contributing to a better understanding of model reasoning and inspiring the development of more effective reasoning models.

## Acknowledgment

This work was conducted while Zhenyu Zhang was a Student Researcher at Google DeepMind.

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

# A. Theoretical Justification

*Proof.* **1) Support identifiability by thresholding.** For $j \in S$,

$$|\langle h, w_j \rangle| = \left| \sum_{i \in S} a_i \langle w_i, w_j \rangle + \langle \varepsilon, w_j \rangle \right|$$

$$\geq |a_j| - \sum_{i \in S \setminus \{j\}} |a_i| \, |\langle w_i, w_j \rangle| - \|\varepsilon\|_2 \ \geq \ \alpha - (k-1)\mu\,\alpha - \sigma.$$

For $j \notin S$,

$$|\langle h, w_j \rangle| \ \leq \ \sum_{i \in S} |a_i| \, |\langle w_i, w_j \rangle| + \|\varepsilon\|_2 \ \leq \ k\mu\,\alpha + \sigma.$$

Hence any $\tau$ in the stated interval yields exact recovery $S^\star(h) = S$. The separation margin is

$$\gamma \ := \ \alpha\big(1 - (2k-1)\mu\big) - 2\sigma \ > \ 0.$$

**2) Stable coefficient recovery on the true support.** Condition on $S$: $h = W_S a_S + \varepsilon$. Then

$$\hat{a}_S - a_S \ = \ (W_S^\top W_S)^{-1} W_S^\top \varepsilon \ = \ W_S^+ \varepsilon,$$

so

$$\|\hat{a}_S - a_S\|_2 \ \leq \ \|W_S^+\|_{\mathrm{op}} \|\varepsilon\|_2 \ = \ \frac{\|\varepsilon\|_2}{\sigma_{\min}(W_S)}.$$

By Gershgorin, $\lambda_{\min}(W_S^\top W_S) \geq 1 - (k-1)\mu$, hence $\sigma_{\min}(W_S) \geq \sqrt{1 - (k-1)\mu}$ and

$$\|\hat{a}_S - a_S\|_2 \ \leq \ \frac{\sigma}{\sqrt{1 - (k-1)\mu}}.$$

**3) Realizability of threshold+LS by a one-hidden-layer ReLU encoder.** The map $h \mapsto S^\star(h)$ is determined by finitely many half-space tests $\{\pm\langle h, w_j \rangle > \tau\}$. On each polyhedral region with fixed $S$, the map $h \mapsto \hat{a}_S$ is affine. Thus $T(h)$ is piecewise affine. Because Step 1 provides a margin $\gamma > 0$, $T$ can be uniformly approximated on the $\gamma/2$-interiors of these regions by a one-hidden-layer ReLU network, with the excluded boundary strip having vanishing probability as $\sigma \to 0$. Hence there exist encoder parameters whose output $z$ coincides with $T(h)$ up to arbitrarily small error while preserving the sparsity pattern $S^\star(h)$.

**4) Population objective separates true and false decoders.** For any $W'$, define

$$\eta_i \ := \ \mathrm{dist}\big(w_i, \mathrm{span}(W')\big).$$

*(a) Span mismatch.* If some $\eta_i > 0$, then on $\mathcal{E}_i(\alpha, \beta)$ we can write $h = a_i w_i + r + \varepsilon$ with $\|r\|_2 \leq \beta$, and by Step 1 the index $i$ is selected. Any reconstruction using $W'$ must approximate $a_i w_i$ from $\mathrm{span}(W')$, incurring error at least $(\alpha - \beta)\eta_i$; hence for some absolute $c > 0$,

$$\|h - \phi_{W'}(h)\|_2^2 \ \geq \ (\alpha - \beta)^2 \eta_i^2 \ - \ c(\beta^2 + \sigma^2).$$

Taking expectations and summing over $i$ yields a strictly positive gap $J(W') \geq \min_{\Pi, D \succ 0} J(W\Pi D) + \delta_{\mathrm{span}} > 0$ for small enough $\beta, \sigma$.

*(b) Same-span misalignment.* Suppose $\mathrm{span}(W') = \mathrm{span}(W)$ but $W' \neq W\Pi D$. Then some $w_i$ is not colinear with any column of $W'$. On $\mathcal{E}_i(\alpha, \beta)$, reconstructing $a_i w_i$ with $W'$ requires combining multiple misaligned columns, causing an oblique-projection loss bounded below by

$$\|a_i w_i - W'T(h)\|_2^2 \ \geq \ c_i(\alpha - \beta)^2 \ - \ c(\beta^2 + \sigma^2),$$

where $c_i > 0$ depends only on the minimal principal angle between $w_i$ and the columns of $W'$. Averaging gives $J(W') \geq \min_{\Pi, D \succ 0} J(W\Pi D) + \delta_{\mathrm{align}} > 0$ for small enough $\beta, \sigma$.

Combining (a) and (b), for any $W' \notin \{W\Pi D\}$ there exists $\delta > 0$ such that

$$J(W') \geq \min_{\Pi, D \succ 0} J(W\Pi D) + \delta.$$

Note that the $\ell_0$ term does not diminish this gap; for $\lambda > 0$ it can only increase it because $T(h)$ is fixed.

**5) From population to empirical risk.** Restrict $W'$ to a compact set $\mathcal{W}$ (e.g., each column norm in $[c, C]$). The map $T$ is piecewise-Lipschitz with finitely many linear regions and $\|T(h)\|_0 \leq k$; hence the class

$$\mathcal{F} := \big\{ h \mapsto \|h - W'T(h)\|_2^2 + \lambda\|T(h)\|_0 \; : \; W' \in \mathcal{W} \big\}$$

has finite pseudo-dimension and satisfies a uniform law of large numbers:

$$\sup_{W' \in \mathcal{W}} \big| J_N(W') - J(W') \big| \xrightarrow[N \to \infty]{\text{a.s.}} 0.$$

Therefore empirical local minimizers converge to the population minimizers, which by Step 4 equal $\{W\Pi D\}$. $\qquad\square$

## B. Empirical Validation of Sparsity and Incoherence Assumptions

Furthermore, we empirically validate the key assumptions underlying the above theoretical analysis: (i) Incoherence, defined as $\max_{i \neq j} \frac{|\langle w_i, w_j \rangle|}{|w_i||w_j|} \leq \mu < 1$, and (ii) Sparsity, characterized by $k < c/\mu$ for a universal constant $c$.

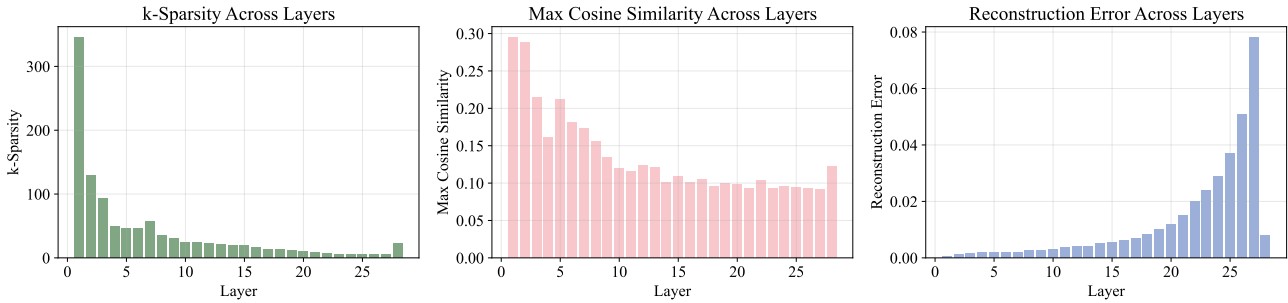

*Figure 10.* Results on Sparsity and Incoherence Assumptions.

Since the ground-truth dictionary $W$ in the theorem 4.1 is not directly observable, we use the reconstructed activations from the sparse autoencoder for evaluation. Given the consistently low reconstruction error in Figure 10, this serves as a reliable proxy. We observe that the maximum cosine similarity among recovered dictionary vectors is clearly bounded by a valid value, and the sparsity level k (defined as the number of active dictionary vectors used to reconstruct each activation) is also well within the theoretical bound. These results provide empirical support for the assumptions underlying our theoretical analysis.

## C. Annotation Details for Reasoning Behaviors

We analyze the consistency of LLM annotation results in Section 4.3, using the following prompt to query GPT-5, GPT-4o, and Claude Sonnet 4.5 for classification.

```
You are a helpful expert that is good at classifying reasoning steps.
You will be given a single reasoning step from a math/logic solution.
Your task is to classify the reasoning step according to the provided taxonomy and
    decision rules.

The available labels are:
(1) reflection: step checking its previous reasoning process and stating its own
    uncertainty.
(2) backtracking: steps that explicitly retract/pivot, proposing an alternative strategy
    to replace the current one.
(3) others: steps that do not fall into the above two categories.
```

```
You must select the class labels based on the above criteria and assign a single class for
    each step.

Your output should be a strict label from the above three options: "reflection", "
    backtracking", or "others".
If you cannot determine the label, please assign "others".

Now, please classify the following reasoning step delimited by triple backticks, according
    to the taxonomy and decision rules provided in the system prompt.
Reasoning Step: ```{text}'''
```

*Table 3.* Keywords set for annotation of reasoning steps.

| Reflection | Wait, verify, make sure, hold on, think again, 's correct, 's incorrect |
| | Let me check, seems right |
| **Backtracking** | Alternatively, think differently, another way, another approach, another method, another solution, |
| | another strategy, another technique |

## D. SAE Reveals the Underlying Geometry of Response Length.

We manually split the step-level activations $\{h_i^l\}$ (where $l$ denotes the layer index) into two categories: *short responses*, with sequence length less than one thousand tokens, and *long responses*, with sequence length exceeding eight thousand tokens. Following the same procedure, we forward the activations from each category into the SAE and highlight the most active columns.

**Structures aligned with response length emerge in the SAE column space.** Figure 11 presents UMAP visualizations of SAE decoder columns across layers, revealing two distinct clusters that align with response length. Columns associated with long responses form a more diverse and dispersed cluster, while those linked to short responses appear compact. This separation is weak in early layers but becomes increasingly clear and stable in mid-to-late layers, peaking just before the output stage. These results suggest that SAE representations progressively encode structural signals about response length alongside behavior-specific features. Also, the normalized silhouette scores across layers, based on response length clustering, are reported in Figure 12. We observe that the early layers exhibit weak separation with respect to response length, while the mid-to-late layers demonstrate stronger alignment with this structural property.

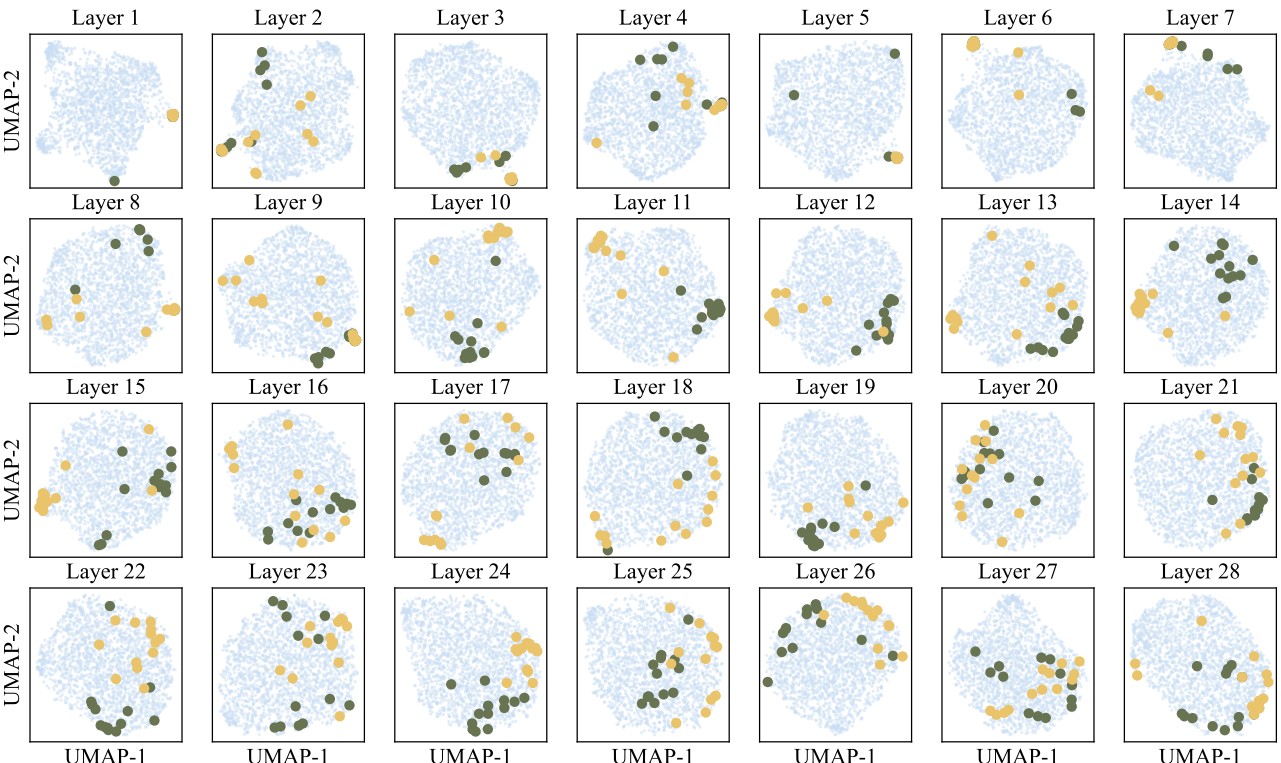

*Figure 11.* Visualization of the SAE decoder columns. From left to right, we show the raw SAE decoder columns and the corresponding results with human-defined behaviors highlighted (green/yellow dots represents the columns related with short and long responses, respectively). Results are obtained from DeepSeek-R1-1.5B using MATH-500 training samples.

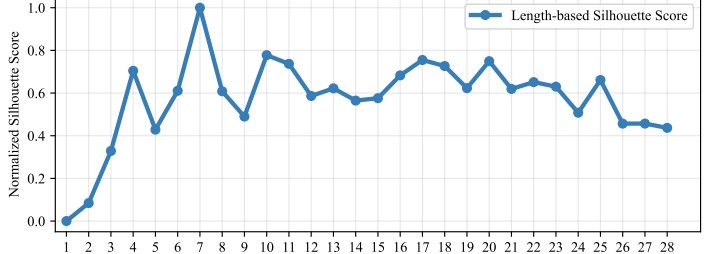

*Figure 12.* Normalized Silhouette scores across different layers of R1-1.5B.

