# OpenReview forum: "Fantastic Reasoning Behaviors and Where to Find Them: Unsupervised Discovery of the Reasoning Process"
_ICML.cc/2026/Conference — ICML 2026 regular_

### Official Review · Reviewer_imgp · 2026-02-25

**Soundness:** 3
**Presentation:** 3
**Significance:** 3
**Originality:** 3
**Overall Recommendation:** 4
**Confidence:** 4

**Summary:**

This paper proposes an unsupervised framework, RISE, which utilizes Sparse Autoencoders (SAE) to obtain sparse representations of sentence-level hidden states, enabling the discovery and manipulation of specific reasoning behaviors without requiring predefined labels. The authors present UMAP visualizations for the "reflection" and "backtracking" behaviors and confirm the high similarity of representations for similar behaviors through Silhouette scores. Furthermore, the corresponding expressive features can be used to manipulate and increase or decrease the targeted behaviors, providing further validation of the causal relationship. The authors also demonstrate representations for concepts such as "response length" and "confidence".

**Compliance With Llm Reviewing Policy:**

Affirmed.

**Final Justification:**

I believe the authors have addressed most of the concerns in their rebuttal. However, the limitation regarding generalization still exists, despite the provided theoretical guarantees and the authors' promise to include them in the updated version. Taking everything into consideration, I will maintain my positive score.

**Key Questions For Authors:**

1.	Why are the expressions for reflection, backtracking, and confidence adjacent? Confidence here corresponds to low entropy, meaning less backtracking and reflection, right? Similar to long and short responses, shouldn’t expressions for high and low confidence be separate?
2.	Are the results sensitive to segmentation? How about reasoning language style? For segmentations other than “\n\n”, such as by sentences or semantics, would the results vary significantly?
3.	It seems that details about hyper-parameter for inference settings are missing.
4.	Can representations corresponding to correct answers (similar to long vs. short responses) be obtained and manipulated to improve performance? Is there a representational difference between tool-use reasoning and pure-text reasoning?

**Limitations:**

Although SAE can acquire representations in an unsupervised manner, the discovery of new concepts still requires definition by humans or models, which could be a potential limitation.

**Strengths And Weaknesses:**

Strengths:

1.	RISE is an unsupervised framework that does not rely on pre-defined concepts, making it well-suited for complex reasoning scenarios.
2.	It is interesting that the framework can not only extract concepts but also directly manipulate reasoning patterns using the corresponding vectors, even for ambiguous concepts like "confidence".
3.	The theoretical results provide a guarantee for the plausibility of SAE.

Weaknesses:

1.	The results may be influenced by the sentence-splitting method; segmentation based on "\n\n" may not be applicable to all models. Moreover, sentence-level segmentation may not fully capture the diversity of reasoning patterns. For instance, the definition of "backtracking" as "proposing an alternative strategy to replace the current one" could be further divided into "reverting to a previously considered line of thought" and "proposing a novel perspective". Sentence-level segmentation may fail to reflect such nuances.
2.	The results may be limited to the distinctive reasoning patterns of the Qwen series of models, as R1-1.5b is also distilled with a Qwen base model. There is a lack of results on models incorporating diverse reasoning patterns.
3.	Practicality is questionable: although RISE can obtain representations in an unsupervised manner, the acquisition of concepts still requires supervision (e.g., annotation by GPT-5 or confidence defined based on entropy). It would be more beneficial if the authors could showcase a broader range of interesting concepts.

---

> ### Author Rebuttal · Authors · 2026-03-31
>
> We sincerely thank Reviewer imgp for the time and effort devoted to reviewing our work. Pointwise responses are provided below.
>
> **[Q1: Sentence-splitting based on "\n\n" may not generalize]** We agree that finer-grained segmentation could capture deeper nuances. We chose \n\n because modern RL reasoning models (like R1) naturally organize their internal "thoughts" into distinct, structured paragraphs separated by this delimiter. Sub-sentence segmentation risks fracturing coherent thoughts, making it harder for the SAE to capture macro-behaviors. We will add a limitation noting that exploring variable-length or semantic segmentation is a promising area for future work.
>
> **[Q2: Results may be limited to Qwen-series models]** We’d like to clarify that our two models, while both distilled, inherit reasoning representations from fundamentally different RL-trained teachers and differ meaningfully in scale (1.5B vs. 8B). R1-1.5B is distilled from DeepSeek-R1, a model trained with large-scale RL on reasoning tasks. Qwen3-8B is distilled from Qwen3 flagship models, which themselves undergo a full four-stage RL post-training pipeline including reasoning-based RL and general RL. The consistent findings across these two models thus provide meaningful evidence beyond a single model family.
>
> **[Q3: Concept acquisition still requires supervision]** We respectfully clarify that supervision is only used for post-hoc validation of the SAE geometry (Section 4.3), not for training RISE itself. The SAE is trained entirely without labels. The GPT-5 annotation is used solely to verify that the unsupervisedly learned geometry aligns with human-interpretable concepts. The confidence vector is identified purely from a model-internal signal (entropy minimization, Equation 7), requiring no natural language definition whatsoever. This is precisely the point we emphasize: RISE can surface concepts like confidence that are difficult to define in text space. Fully disentangling the SAE geometry to systematically discover and catalog all latent reasoning concepts is an important and interesting open question, which we identify as a promising future direction. The present work is intended as a pivot study demonstrating the feasibility and potential of this approach.
>
> **[Q4: Why are reflection, backtracking, and confidence adjacent]** The adjacency of confidence vectors to reflection and backtracking regions (Figure 8a) is itself a meaningful finding: it suggests that confidence is geometrically related to these metacognitive behaviors, which is consistent with our observation that applying the confidence vector reduces reflection and backtracking steps (Table 2). Regarding high vs. low confidence — our entropy-based identification naturally finds the low-entropy (high confidence) direction. The opposite direction would correspond to high entropy (low confidence), which geometrically would point away from this region.
>
> **[Q5: Are results sensitive to segmentation?]** The "\n\n" delimiter is the native step boundary in the model responses, providing a scalable and consistent way to collect a large number of step-level representations. Regarding sensitivity to alternative segmentation: the key insight from Theorem 4.1 is that SAE recovery does not require each representation to encode exactly one concept (k=1) — it only requires that representations admit a k-sparse decomposition over the latent dictionary for some k≥1. Different segmentation granularities essentially change how many concepts are merged into a single representation. Crucially, as long as the k-sparsity condition is satisfied, Theorem 4.1 guarantees recovery of the underlying behavior dictionary regardless of granularity. Figure 10 (left) empirically confirms that the sparsity level k is small but not strictly 1 across layers, indicating there is sufficient buffer to accommodate different segmentation choices. We therefore expect the results to be robust to reasonable alternative segmentation strategies.
>
> **[Q6: Details about inference hyperparameters]** We use the default settings: temperature = 0.6 and top-p = 0.95, which we will include in the updated version.
>
> **[Q7: Can representations for correct answers be obtained? Is there a difference between tool-use and pure-text reasoning?]**
> Identifying a "correctness vector" is a natural extension of RISE that could provide a principled way to steer reasoning toward correct trajectories at inference time. The key challenge lies in defining a differentiable mathematical proxy for correctness to score SAE concepts. One potential approach is to measure accuracy over grouped responses per concept vector, though this is computationally intensive. Similarly, the distinction between tool-use and pure-text reasoning representations is an interesting open question, as tool-use involves structurally different reasoning steps that may occupy distinct regions of the SAE geometry. We will explore both directions in future work.

---

> > ### Author Rebuttal · Reviewer_imgp · 2026-04-04
> >
> > I thank the authors for their rebuttal! I believe that Q3, Q5, Q6, and Q7 have been fully addressed. However, regarding Q4, I still have a doubt: theoretically, 'reflection' and 'backtracking' should correspond to 'low confidence', whereas the 'confidence vector' identified in the paper represents 'high confidence'. Does this imply that 'reflection' and 'backtracking' should be geometrically distant from the high-confidence direction, which seems to contradict the actual observations?
> >
> > Furthermore, I believe the limitations in Q1 and Q2 still persist, as reasoning models such as gpt-oss may exhibit reasoning patterns that are fundamentally different from the models considered in this study. Taking these factors into account, I will maintain my current score for now. If the authors provide more substantial evidence to resolve these points, I would be open to reconsidering my evaluation.

---

> > > ### Author Response · Authors · 2026-04-08
> > >
> > > Thank you for the follow-up questions. We address them below.
> > >
> > > For Q4, we'd like to clarify that the confidence vector is identified based on entropy as defined in Equation 7, where high entropy corresponds to low confidence, meaning the confidence vector is more precisely a "confidence-related" vector that captures the model's uncertainty signal, rather than strictly a "high confidence" vector. This entropy-based signal is intrinsically related to reflection and backtracking, since uncertain states (high entropy) are precisely what triggers these metacognitive behaviors.
> > >
> > > Regarding the generalization of RISE, we agree that evaluating on more diverse models would further strengthen the paper. Due to the limited time during rebuttal, we will include additional models in the updated version. Beyond this, we would like to highlight that RISE's generalization is grounded both theoretically and empirically. Theoretically, by Theorem 4.1, the SAE reconstruction does not rely on the specific activation geometry of any particular model. it recovers the underlying latent behavior dictionary as long as the linear representation hypothesis holds, which has been empirically validated at scale across diverse model families by prior works. Empirically, The consistent effectiveness of RISE on R1-1.5B and Qwen-8B provides meaningful evidence for generalization across different models.

---

### Official Review · Reviewer_UjbY · 2026-02-28

**Soundness:** 3
**Presentation:** 3
**Significance:** 2
**Originality:** 2
**Overall Recommendation:** 4
**Confidence:** 4

**Summary:**

This paper uses the Sparse AutoEncoder method to find steering vectors that corresponds to certain reasoning process, such as reflection and backtracking. These vectors are carefully examined on correlation and causality to the connected concepts. The authors also provide experiments to show its potential on improving reasoning capabilities of LLMs.

**Compliance With Llm Reviewing Policy:**

Affirmed.

**Final Justification:**

My concerns for further improving the paper are not fully addressed. During the rebuttal, the authors admit these weaknesses exist and explain why they occur, but they stop short of fundamentally resolving the issues. Though I still believe the paper deserves a weak accept.

**Key Questions For Authors:**

My concerns have been provided in the weaknesses part.

**Limitations:**

Yes.

**Strengths And Weaknesses:**

**Strengths**
1.	Standard SAE-style paper with detailed examination on the interpretable vectors. There’s little to criticize.
2.	The obtained vectors (confidence vectors) are shown to be effective to enhance the reasoning process in both math and commonsense QA scenarios.
3.	The paper is easy to understand and easy to follow.
**Weaknesses**
1.	The analytical procedure is so standardized to the point that it lacks novelty. Given that SAE has already been shown to identify meaningful steering vectors across various domains, discovering vectors related to reasoning is not particularly innovative. For example, several studies long CoT by manually inserting “wait” tokens, which substantially overlaps with the reflection vectors identified by the authors.
2.	In Section 5, the paper combine top-3 confidence vectors to improve reasoning accuracy, which seems not aligned well with the Incoherence Assumptions in the paper. Does it means there are several vectors that shares the same interpretable concept while their directions differ? If my understanding is correct, then there may be some concern about the uniqueness of the obtained steering vectors.
3.	Are reflection and backtracking vectors useful to enhance reasoning process? Why they are not combined with the confidence vectors? Also, there may be more other vectors that can enhance reasoning capabilities of LLMs.
4.	Theorem 4.1 relies on the assumption that hidden representations at delimiter tokens follows Eq. (3), which is not so natural. Are there any support on this assumption?

---

> ### Author Rebuttal · Authors · 2026-03-31
>
> Many thanks to Reviewer UjbY for the detailed and constructive suggestions. Pointwise responses are provided in the following:
>
> **[Q1: Lack of novelty given prior SAE work]** We respectfully clarify that the novelty of RISE is not merely in applying SAEs to find steering vectors, but in three specific contributions: (1) the unsupervised discovery of reasoning behaviors without requiring human-defined labels, in contrast to prior work that constructs contrastive datasets from predefined concepts; (2) the theoretical guarantee in Theorem 4.1 that SAE decoder columns recover the ground-truth latent behavior dictionary; and (3) the discovery of behaviors, such as the confidence vector, that are difficult to define at the word level, going strictly beyond what supervised approaches can find.
>
> Regarding the overlap with "wait" token insertion: while inserting "wait" tokens empirically prolongs reasoning, it operates at the token level and cannot distinguish which internal behavior is being activated. RISE instead identifies the precise geometric direction in activation space corresponding to reflection, enabling targeted control that token-level methods cannot provide.
>
>
> **[Q2: Combining top-3 confidence vectors seems inconsistent with the Incoherence Assumption]** This is an insightful question. The incoherence assumption requires that distinct dictionary vectors are not collinear. It does not require that each semantic concept is represented by exactly one vector. In practice, a high-dimensional concept such as "confidence" may be encoded across multiple partially overlapping directions, each capturing a different facet (e.g., confidence in arithmetic steps vs. confidence in logical conclusions). The top-3 confidence vectors we combine are selected by their entropy-reduction scores (Equation 7) and their combination coefficients $\alpha$ are learned per-sample at test time, making the steering sample-dependent rather than fixed. The incoherence assumption governs recovery of the dictionary, not the cardinality of vectors per concept. So combining multiple confidence vectors is theoretically consistent with the framework.
>
>
> **[Q3: Are reflection and backtracking vectors useful for enhancing reasoning? Why not combine them with confidence vectors?]** Good question. We found that suppressing reflection/backtracking naturally pushed the model into a higher-confidence state, which improved both accuracy and speed (as seen in our confidence vector interventions). Artificially amplifying reflection can sometimes cause the model to overthink or hallucinate errors in otherwise correct logic. However, combining these vectors to dynamically control the compute-to-accuracy tradeoff is an excellent direction that we will highlight in the revision.
>
> We acknowledge that additional concept vectors beyond those identified in this work may also enhance LLM reasoning capabilities. The primary contribution of RISE is to propose an unsupervised, scalable framework for discovering such behaviorally relevant representations, rather than providing an exhaustive enumeration of all reasoning-relevant concepts. Precisely characterizing which concepts do or do not enhance reasoning is a compelling direction that would require, among other things, a formal mathematical definition of reasoning capability, a scoring mechanism to evaluate each concept vector in the SAE feature space, and causal validation of their downstream effects. We consider this an inspiring avenue for future work, and leave it as an open problem beyond the scope of the current paper.
>
>
> **[Q4: The assumption in Theorem 4.1 that hidden representations follow Eq. (3) is not natural — is there empirical support?]**
> We appreciate this careful reading. The generative model in Eq. (3) — that activations are sparse linear combinations of a latent behavior dictionary plus bounded noise — is the standard assumption underlying the linear representation hypothesis[1,2], which has been empirically validated across a wide range of models and scales. More directly, we provide empirical validation of the two key conditions in Appendix B (Figure 10): the maximum cosine similarity among recovered dictionary vectors (incoherence) is consistently bounded well below 1 across all layers, and the sparsity level k remains well within the assumption.
>
>
> [1] The Linear Representation Hypothesis and the Geometry of Large Language Models
>
> [2] The Dark Matter of Neural Networks?

---

> > ### Author Rebuttal · Reviewer_UjbY · 2026-04-01
> >
> > I have received point to point response from the author. In their response, the author admits these weaknesses exist and explains why they occur, but they stop short of fundamentally resolving the issues (I understand it's hard to address this just in a week). Thus, I will keep my score to support weak accept.

---

### Official Review · Reviewer_ty6s · 2026-03-13

**Soundness:** 2
**Presentation:** 2
**Significance:** 2
**Originality:** 2
**Overall Recommendation:** 3
**Confidence:** 4

**Summary:**

The paper introduces RISE (Reasoning behavior Interpretability via Sparse auto-Encoder), an unsupervised framework for identifying and controlling reasoning mechanisms in large language models. The authors train sparse auto-encoders (SAEs) on sentence-level activations from chain-of-thought traces to uncover reasoning vectors, linear directions in the model’s latent space associated with interpretable behaviors such as reflection and backtracking. They show that these behaviors occupy distinct regions in activation space, and that intervening on the corresponding vectors can causally amplify or suppress specific reasoning traits at inference time without retraining the model. RISE also discovers previously unidentified behaviors, such as confidence-related reasoning, demonstrating that unsupervised latent feature discovery can both interpret and steer LLM reasoning.

**Compliance With Llm Reviewing Policy:**

Affirmed.

**Final Justification:**

1 sensitivity to Step Segmentation: This work treats individual sentences as reasoning steps. However, sentence boundaries are often arbitrary and can be easily restructured. Would the pattern remain consistent if the reasoning trace were re-segmented or if the sequence of sentences were permuted?

2 Discovering new vectors: How do you validate that confidence vectors actually measure what you claim rather than some correlated property? How do you verify they are fundamentally disentangled from other reasoning vectors? The understanding of this part is limited and over-claimed. Classifying reasong traces with literal features and entropies are not conflicting and may be overlapping. Currently, discovering new vectors seems over-claimed and lacks verification.

3 The 'other' category: I still find this not rigorous. While past works have already classified reasoning traces into more types based on literal meanings, this work only looks at two behaviors that have been studied before, and then somehow there is a category called ``other''. This is especially unacceptable when authors claim their work can discover new reasoning behaviors.

4 geometric interpretation: I am still concerned with the separability or more fundamentally, the interpretation of different reasoning races. Likely, they pretty much share something in the latent space, and it implies the literal classification based on the meaning is not reliable and may not reveal the fundamental way how LLMs treat and organize reasoning behaviours in the latent space. Stronger analysis is needed to understand what LLMs truly encode.

Overall, I still share the same concern as other reviewers that this work has limited novelty and contribution, and the generalizability is questionable.

**Key Questions For Authors:**

what are ``other'' reasoning behaviors?

How is the reasoning performance influenced when intervening with SAE?

**Limitations:**

yes

**Strengths And Weaknesses:**

Strength:

The proposed SAE framework seems to capture features of different reasoning behaviors, and can be used to control reasoning behaviors.
The learned confidence vector can help improve reasoning accuracy and reduce thought time.
The authors show visual overlap between confidence vector and vectors for reflection and backtracking.

Weakness:

Only math data and one small thinking model are employed.
The generalization of confidence vector is unclear. The learning seems to be post-hoc to a specific dataset.
Figure 2 does not show very separable patterns. The work seems to primarily focus on backtracking and reflection, which is very limited. It would be interesting to analyze the clustering and understand how LLMs themselves define different reasoning behaviors internally.

The authors should discuss the following two works, which are very close.
Bogdan, Paul C., et al. "Thought Anchors: Which LLM Reasoning Steps Matter?." arXiv preprint arXiv:2506.19143 (2025).
Venhoff, Constantin, et al. "Understanding reasoning in thinking language models via steering vectors." arXiv preprint arXiv:2506.18167 (2025).

---

> ### Author Rebuttal · Authors · 2026-03-31
>
> We sincerely thank Reviewer ty6s for the time and effort in reviewing our work. The suggestions are really helpful to further strengthen our work. To address the concerns, we provide pointwise responses in the following:
>
> **[Q1: Limited evaluation tasks and model scales]** Thanks. We first clarify that our experiments extend well beyond math: Tables 1 and 2 demonstrate consistent generalization to commonsense and logical reasoning (GPQA-Diamond, KnowLogic), and Figure 6 shows consistent results across model scales from 1.5B to 8B. Scaling to 70B+ models is currently limited by the associated computational overhead. However, the theoretical foundation of RISE is the linear representation hypothesis, which has been widely validated at a much larger scale by prior works[1,2]. Also, our SAE training objective is model-agnostic: by Theorem 4.1, it recovers the original geometric structure of hidden representations regardless of how they were produced. We therefore believe the current experiments provide satisfactory evidence for the effectiveness and generality of our method, and we will add an explicit discussion of the generalization question in the updated version.
>
> **[Q2: Generalization of the confidence vector]** The confidence vector is not post-hoc to a specific dataset. It is identified once using the MATH500 training set (Line 317), then fixed and applied without modification to all intervention experiments: AIME25 (same-domain generalization) and GPQA-Diamond and KnowLogic (cross-domain generalization). The consistent results across all three settings confirm the confidence vector's generalization.
>
> **[Q3: Figure 2 does not show separable patterns; analysis limited to reflection and backtracking]** Thanks for the question. While 2D UMAP visually compresses high-dimensional structure, the quantitative Silhouette scores in Figure 4 confirm statistical separability, particularly in mid-to-late layers. Reflection and backtracking are not chosen arbitrarily — they are the most prominent metacognitive behaviors introduced by current reasoning models and are the primary focus of prior work [3, 4, 5].
>
> More broadly, the confidence vector is introduced precisely to move beyond human-defined behavior labels. While reflection and backtracking require natural language definitions prior to discovery, the confidence vector is defined entirely by a model-internal signal (entropy), enabling the discovery of behaviors that may be difficult or impossible to express in natural language. Regarding pure clustering analysis: such an approach requires predetermining the number of clusters and subsequently interpreting the centroid vector of each cluster. The predefined number of clusters is a sensitive hyperparameter and the resulting clusters are prone to noise, and there is no guarantee that cluster centroids correspond to human-interpretable concepts, making verification inherently difficult. The confidence vector approach avoids these issues by grounding discovery in a principled and well-defined model-internal signal, serving as a pivot study that demonstrates the feasibility of understanding internal reasoning geometry beyond the constraints of natural language supervision.
>
> **[Q4: Discussion of related works]** We thank the reviewer for pointing out [6]. Our work targets a different research question: rather than identifying which reasoning steps matter for the final answer, we focus on understanding the geometry of reasoning behaviors in activation space, providing a broader map of the internal reasoning structure. Regarding [7], their key limitation is requiring human-defined word-level behavior labels before finding steering vectors — which may miss behaviors that models use but that are difficult to express in natural language. This is precisely our motivation for an unsupervised approach, and the confidence vector is a concrete demonstration that such beyond-language behaviors exist and can be discovered. We will include the discussion in the updated manuscript.
>
> **[Q5: What are "other" reasoning behaviors?]** The "other" category encompasses all reasoning steps that are neither reflective nor backtracking.
>
> **[Q6: How is reasoning performance influenced when intervening with SAE?]** Thanks for the question. As reported in Figure 9, intervening with the confidence vector improves reasoning accuracy on MATH500 and AIME 2025 by up to 4.66 points while reducing token usage by 13.69%,
>
> [1] Scaling Monosemanticity: Extracting Interpretable Features from Claude 3 Sonnet. Transformer Circuits Thread.
>
> [2] Scaling and evaluating sparse autoencoders
>
> [3] SEAL: Steerable Reasoning Calibration of Large Language Models for Free
>
> [4] Do NOT Think That Much for 2+3=? On the Overthinking of o1-Like LLMs
>
> [5] Thoughts are all over the place: On the underthinking of o1-like llms.
>
> [6] Thought Anchors: Which LLM Reasoning Steps Matter?
>
> [7] Understanding reasoning in thinking language models via steering vectors.

---

> > ### Author Rebuttal · Reviewer_ty6s · 2026-04-03
> >
> > I appreciate the authors' rebuttal. This work does have merits; however, I think this work has some weakness that is hard to resolve and require more fundamental analysis.

---

> > > ### Author Response · Authors · 2026-04-03
> > >
> > > Many thanks to Reviewer ty6s for the continued engagement. We would greatly appreciate it if the reviewer could elaborate on the specific weaknesses that remain unresolved, as we are very willing to discuss further. Thank you!

---

### Official Review · Reviewer_HYs2 · 2026-03-15

**Soundness:** 3
**Presentation:** 3
**Significance:** 3
**Originality:** 3
**Overall Recommendation:** 4
**Confidence:** 4

**Summary:**

This paper proposes an unsupervised framework that uses sparse autoencoders to investigate internal reasoning vectors in large language models. The paper further validates through intervention experiments that the vectors identified by the SAE can be used to controllably enhance or suppress specific reasoning behaviors. Additionally, they uncover novel vectors capable of modulating the model’s response confidence.

**Compliance With Llm Reviewing Policy:**

Affirmed.

**Final Justification:**

I kind of agree with Reviewer ty6s regarding the limited evaluation, but I think this is a common limitation of most current interpretability papers: they are still largely at the stage of theoretical exploration and have yet to demonstrate substantial practical impact. Nevertheless, I still believe these papers are important for helping us understand model behavior, especially as models become increasingly capable. Compared with prior work, this paper has two notable strengths. First, it identifies novel vectors beyond those derived from human supervision, thereby extending the scope of earlier work on reasoning vectors and offering new inspiration to the community. Second, by intervening on the confidence vector, the authors achieve relatively substantial accuracy gains on AIME25 and MATH500 while also reducing token usage, which significantly enhances the paper’s practical relevance. Therefore, I think this paper deserves a weak accept.

**Key Questions For Authors:**

See weaknesses

**Limitations:**

yes

**Strengths And Weaknesses:**

Strengths:
1. The paper is well written and easy to follow.
2. The paper identifies novel vectors beyond those derived from human supervision, extending the scope of prior work on reasoning vectors and providing new inspiration for the community.
3. The paper validates through intervention experiments that the identified reasoning vectors can indeed be used to control specific model behaviors.
4. By intervening on the confidence vector, the model achieves relatively significant accuracy improvements on AIME25 and MATH500 while also reducing token usage, which substantially strengthens the practical significance of the paper.

Weaknesses:
1. Although the paper presents experiments and analysis on DeepSeek-R1-Distill-Qwen-1.5B and Qwen3-8B, both models are distilled and relatively small in scale. Demonstrating that the findings also hold on larger models trained with reinforcement learning would substantially strengthen the paper’s significance and generality. At a minimum, the paper should include a discussion addressing whether the conclusions are likely to generalize to such models.

---

> ### Author Rebuttal · Authors · 2026-03-31
>
> We sincerely thank Reviewer HYs2 for the thoughtful and positive assessment, and for acknowledging the contributions of our work. We find the raised concern constructive and provide the following response to address it.
>
> **[Q1: Generalization to larger RL-trained models.]** Thanks for the good question.
>
> Firstly, The validity of RISE rests on the linear representation hypothesis, which is not specific to distilled or small models. Prior interpretability work has empirically validated this across a wide range of model families and scales [1, 2, 4]. Most directly, [3] successfully extracts high-quality interpretable features via SAEs from Claude 3 Sonnet, which is a large production-scale model trained with RL-based alignment. This result demonstrates that linear semantic structure in representations generalizes to large RL-trained models across different scales.
>
> Then, from Theorem 4.1, the SAE training objective is model-agnostic where SAE training recovers the original geometric structure of hidden representations regardless of how those representations were produced (distillation vs. RL vs. pretraining).
>
> Empirically. Our two models, while both distilled, inherit reasoning representations from fundamentally different RL-trained teachers and differ meaningfully in scale (1.5B vs. 8B). DeepSeek-R1-Distill-Qwen-1.5B is distilled from DeepSeek-R1, a model trained with large-scale RL on reasoning tasks, so its internal representations are strongly shaped by RL-induced reasoning patterns. Qwen3-8B is distilled from Qwen3 flagship models, which themselves undergo a full four-stage RL post-training pipeline including reasoning-based RL and general RL. Our method demonstrates consistent results across these two architecturally distinct models with different teachers, which also provides meaningful evidence for generality beyond any single training recipe.
> We appreciate the reviewer's suggestion to validate on larger fully RL-trained models. Due to the computational cost of collecting step-level activations at scale for SAE training, we believe that experiments are currently beyond the scope of this work. We have updated the camera-ready version to include an explicit discussion of the generalization question across different training recipes.
>
> [1] The Linear Representation Hypothesis and the Geometry of Large Language Models. Kiho Park, Yo Joong Choe, Victor Veitch
>
> [2] Sparse Autoencoders Find Highly Interpretable Features in Language Models.
>
> [3] Scaling Monosemanticity: Extracting Interpretable Features from Claude 3 Sonnet. Transformer Circuits Thread. https://transformer-circuits.pub/2024/scaling-monosemanticity/
>
> [4] Scaling and evaluating sparse autoencoders. Leo Gao, Tom Dupré la Tour, Henk Tillman, Gabriel Goh, Rajan Troll, Alec Radford, Ilya Sutskever, Jan Leike, Jeffrey Wu. ICLR 2025.

---

> > ### Author Rebuttal · Reviewer_HYs2 · 2026-04-01
> >
> > Thanks for the authors’ response. My concerns have been addressed, and I will maintain my positive score.

---

### Decision · Program_Chairs · 2026-04-30

**Decision:**

Accept (regular)

**Comment:**

According to reviews, the major concerns include:
1. this work has limited novelty and contribution, and the generalizability is questionable
2. The uniqueness of the obtained steering vectors is questionable.
3. Sub-sentence segmentation may risks fracturing coherent thoughts. It could be harder for the SAE to capture macro-behaviors.
4. Results may be limited to Qwen-series models.

However, most reviewers gave positive scores. And I read the rebuttals and discussions between the reviewers and authors, most concerns are addressed.